# Protocol for a case–control diagnostic accuracy study to develop diagnostic criteria for psoriasis in children (DIPSOC study): a multicentre study recruiting in UK paediatric dermatology clinics

Esther Burden-Teh,[1] Ruth Murphy,[1] Sonia Gran,[1] Tamar Nijsten,[2] Carolyn Hughes,[1] Kim Suzanne Thomas [1]

¹Centre of Evidence Based Dermatology, School of Medicine, University of Nottingham, Nottingham, UK
²Department of Epidemiology, Erasmus MC, Rotterdam, The Netherlands

**Correspondence to**
Dr Esther Burden-Teh;
esther.burden-teh@nottingham.ac.uk

## ABSTRACT

**Introduction** Diagnosing psoriasis in children can be challenging. Early and accurate diagnosis is important to ensure patients receive psoriasis specific treatment and monitoring. It is recognised that the physical, psychological, quality of life, financial and comorbid burden of psoriasis are significant. The aim of this study is to develop clinical examination and history-based diagnostic criteria for psoriasis in children to help differentiate psoriasis from other scaly inflammatory rashes. The criteria tested in this study were developed through a consensus study with a group of international psoriasis experts (International Psoriasis Council).

**Methods and analysis** Children and young people (<18 years) with psoriasis (cases) and other scaly inflammatory skin diseases (controls) diagnosed by a dermatologist are eligible for recruitment. All participants complete a single research visit including a diagnostic criteria assessment by a trained investigator blinded to the participant's diagnosis. The reference standard of a dermatologist's diagnosis is extracted from the medical record. Sensitivity and specificity of the consensus derived diagnostic criteria will be calculated and the best predictive criteria developed using multivariate logistic regression.

**Ethics and dissemination** Health Regulatory Authority and National Health Service Research Ethics Committee approvals were granted in February 2017 (REC Ref: 17/EM/0035). Dissemination will be guided by stakeholders; patients, children and young people, dermatologists, primary care and paediatric rheumatologists. The aim is to publish the study results in a high-quality peer-reviewed journal, present the findings at international academic meetings and disseminate more widely through social media and working with patient associations.

**Trial registration number** ISRCTN98851260.

## Strengths and limitations of this study

► This is a UK multicentre study recruiting 320 consecutive children and young people in 12 paediatric dermatology departments.

► The trained investigator undertaking the diagnostic criteria assessment is blinded to the participant's reference standard of a dermatologist's diagnosis.

► A case–control study design is likely to overestimate the diagnostic accuracy, but this is an appropriate and feasible study design for a diagnostic criteria development study.

► External validation of the diagnostic criteria will be needed in the setting and population that the criteria will be used.

Organisation (WHO) as a serious non-communicable disease and an area of unmet health need.[3] Although the exact causes for the onset of psoriasis are not fully understood, they originate in a complex interaction between genetic and environmental factors.[4] Psoriasis can affect the face, hands, nails, genitals and flexures, therefore, it is both the extent and the location of disease that are important to patients. It is known that the physical, psychological, quality of life, financial and comorbid burden of psoriasis are significant.[5–11]

Psoriasis can affect people of all ages. However, making the diagnosis in children and young people can be more challenging compared with diagnosing psoriasis in adults. Psoriasis is often under-recognised in this younger age group and may be misdiagnosed as other common red scaly rashes such as eczema, viral exanthems and fungal infections. The clinical features seen in childhood

## INTRODUCTION

Psoriasis is a chronic immune-mediated inflammatory disease of the skin and joints.[1 2] It is recognised by the World Health

disease are often more subtle with thinner plaques, facial involvement and flexural disease in hidden sites normally covered by clothing.[12–14] Therefore, the diagnosis of psoriasis in children and young people may be missed by non-dermatologists.

Epidemiological data are limited, but it is estimated that one-third of adults with psoriasis first develop skin changes in childhood.[15 16] Therefore, early and accurate diagnosis presents an opportunity for early intervention. National Institute for Health and Care Excellence (NICE) recommends all children with suspected psoriasis are referred to a dermatology specialist for assessment and management.[17] This specialist review also includes initiating monitoring for comorbid diseases and assessment for juvenile psoriatic arthritis. Accurate recognition of psoriasis is also important to help paediatric rheumatologists differentiate juvenile idiopathic arthritis into juvenile psoriatic arthritis. This differentiation alters the treatment pathway and likely prognosis for children with juvenile psoriatic arthritis.[18] Ensuring children and young people receive psoriasis specific treatment and monitoring from the onset is important to help minimise the negative long-term consequences of psoriasis, known as cumulative life course impairment.[19]

There are no clinical examination-based diagnostic criteria for psoriasis.[20] Diagnosis in clinical practice currently relies on expert pattern recognition by a trained dermatologist.[21 22] Skin biopsies are not routinely taken, especially in children. Consequently, there are no available diagnostic aids to support non-dermatologists to recognise psoriasis in children. In research studies, the case-definition and eligibility criteria for psoriasis in children are often poorly described, reducing the generalisability and ability to synthesis studies.[23 24]

Improving awareness of psoriasis has been identified as a topic of importance in the Psoriasis and Psoriatic Arthritis Alliance (PAPAA) prioritisation exercise. The recently completed James Lind Alliance Psoriasis Priority Setting Partnership (PSP) identified 10 research priorities in psoriasis that are important to people who have psoriasis, their families and friends, and the healthcare professionals who treat them. The second priority asks 'Does treating psoriasis early (or proactively) reduce the severity of the disease, make it more likely to go into remission, or stop other health conditions developing'.[25] In children and young people, ensuring early and accurate recognition of psoriasis will be a necessary part of answering this question.

An initial eDelphi consensus study has been completed with a group of global clinically active psoriasis experts who are members of the International Psoriasis Council. The group agreed 16 clinical features that are important for the diagnosis of plaque psoriasis in children.[26]

The next step in developing diagnostic criteria for psoriasis in children is to empirically test how well the consensus-derived diagnostic criteria perform and to refine the criteria. The Developing **DI**agnostic criteria for **PSO**riasis in **C**hildren (DIPSOC) study has been designed to

> **Box 1  Sixteen diagnostic features agreed by the International Psoriasis Council to be important for the diagnosis of plaque psoriasis in children (26). Two additional diagnostic features (\*) have also been included that were close to reaching consensus and were emphasised as important in the feedback from experts.**
>
> **Major criteria**
> ► Scaly erythematous plaques on the extensor surfaces of the elbows and knees.
> ► Scaly erythematous plaques on the trunk triggered by a sore throat or other infection.
> ► Raindrop plaques typical of guttate disease on the trunk or limbs.
>
> **Minor criteria**
> ► Scale and erythema in the scalp involving the hairline.
> ► Retroauricular erythema (including behind the earlobes).
> ► Scaly erythema inside the external auditory meatus.
> ► Persistent well-demarcated erythematous scaly rash anywhere on the body.
> ► Fine scaly patches involving the upper thighs and buttocks.
> ► Well-demarcated erythematous rash in the napkin area involving the crural folds.
> ► Persistent erythema in the umbilicus.
> ► Nail pitting.
> ► Onycholysis of the nail(s).
> ► Subungual hyperkeratosis of the nail(s).
> ► Positive family history of psoriasis.
> ► Koebner phenomenon.
> ► Fusiform swelling of a toe or a finger suggestive of dactylitis.
>
> \*Persistent well-demarcated facial rash with fine or absent scale.
> \*Natal cleft erythema and/or skin splitting.

develop a diagnostic tool for identifying childhood psoriasis. The primary objective is to test the diagnostic accuracy (sensitivity and specificity) of the consensus agreed diagnostic criteria (box 1) and develop the best predictive diagnostic criteria using multivariate analysis. DIPSOC is a development study because further validation work is needed before the diagnostic accuracy of the criteria in primary, secondary and research settings are known.

## METHODS
### Study design
DIPSOC is a multicentre case–control diagnostic accuracy study with a nested substudy. The nested substudy is recruiting children and young people with indeterminate psoriasis alongside the main study. The full protocol was lodged on the Centre of Evidence Based Dermatology website prior to the first participant being recruited. This published protocol is based on Protocol 12.10.2017 Final V.1.2. The full protocol is available on the Centre of Evidence Based Dermatology University of Nottingham website www.nottingham.ac.uk/go/dipsoc.

### Primary objective
To test the diagnostic accuracy of the consensus agreed diagnostic criteria for plaque psoriasis in children/young

people and develop the best predictive diagnostic criteria using multivariate analysis.

## Secondary objectives
1. To compare the diagnostic performance of the consensus agreed diagnostic criteria and the best predictive criteria for plaque psoriasis in children and young people.
2. To assess the interobserver variability in the diagnostic criteria assessment.
3. To assess the variability in the reference standard for psoriasis.

## Setting
DIPSOC is recruiting in 12 UK paediatric dermatology outpatient clinics in secondary and tertiary care. This setting is a feasible environment in which the reference standard (dermatologist's diagnosis) can be obtained and the sample size recruited within the time and resources available. A specialist setting is appropriate for a development study, but validation research will be needed to test the performance of the diagnostic criteria in the settings that they are intended to be used (eg, primary care and paediatric rheumatology clinics).

## Participant selection
### Inclusion criteria
Cases and controls are children and young people aged 0 to <18 years, with active skin disease (rash present) at the time of assessment and are able to consent or have a parent/guardian willing to give consent.

Cases have a confirmed diagnosis of plaque psoriasis by a dermatologist. Plaque psoriasis has been used as a broad term to include all subtypes and presentations of psoriasis where plaques are the main feature. For example, chronic plaque psoriasis, guttate psoriasis, scalp psoriasis and flexural psoriasis are included but purely nail psoriasis or juvenile psoriatic arthritis without skin involvement are excluded. The decision to include guttate psoriasis under the broad description of plaque psoriasis was agreed with the International Psoriasis Council.

Controls have a confirmed diagnosis of a scaly inflammatory rash (excluding psoriasis or indeterminate psoriasis) by a dermatologist. Skin conditions that may be included in the control population are eczema (atopic dermatitis), pityriasis rubra pilaris, pityriasis rosea, ichthyosis, mycosis fungoides, Gianotti-Crosti and tinea corporis. These conditions are not an exhaustive list and the decision as to whether a participant's skin disease meets the eligibility criteria is made by the patient's dermatologist.

### Exclusion criteria
Children or young people with pustular psoriasis, erythrodermic psoriasis or do not have a dermatologist's diagnosis.

## Index test
A diagnostic criteria assessment looking for the presence or absence of each of the diagnostic features (box 1).

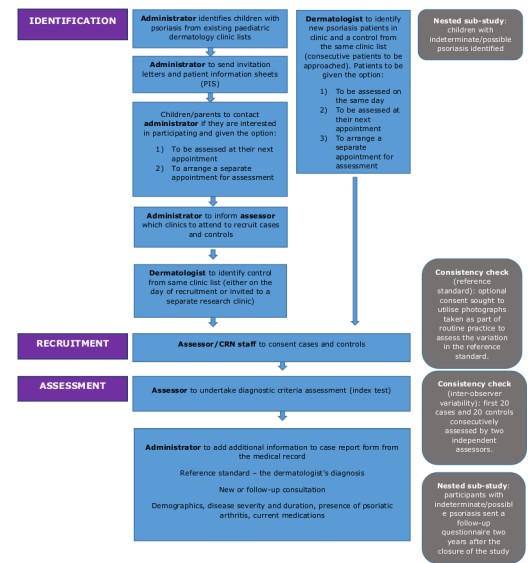

**Figure 1** Developing DIagnostic criteria for PSOriasis in children study flow. CRN, Clinical Research Network.

Using the same assessment, the index test is divided into index test 1 and index test 2.

### Index test 1
The international eDelphi consensus study agreed 16 diagnostic features of childhood psoriasis and separated them into major and minor criteria. In the consensus study, a scoring algorithm was proposed where the presence of one or more major criteria or three of more minor criteria would support a diagnosis of psoriasis. Together these 16 diagnostic features and the scoring algorithm form index test 1.[26]

### Index test 2
Two additional features were close to reaching consensus and were emphasised as important in the feedback from the expert participants. These 18 items will be used to create the best predictive criteria using multivariate analysis. The best predictive criteria form index test 2.

### Reference test
A dermatologist's diagnosis as recorded in the participant's medical record. The diagnosis is a clinical diagnosis and may include, but does not require, a skin biopsy.

### Study flow
The study flow is depicted in figure 1. Children and young people who meet the eligibility criteria to be a case or control are approached by their usual dermatology team. They are invited to attend a research visit on the same day, at their next consultation or at a separate research visit. All consecutive psoriasis patients are being approached and consecutive control patients when a case is identified. Cases identified from existing medical records are approached by letter from their usual dermatology team.

After informed consent has been taken, all participants undergo the same research visit. The visit comprises demographic questions, quality of life questionnaires (for those

> **Box 2    A summary of the data variables collected in the DIPSOC study**
>
> **Research visit**
> ► Demographic information: age, sex, ethnicity, household occupation.
> ► Diagnostic criteria assessment: presence and absence of each of the 18 diagnostic features (index test).
> ► Clinical experience of the diagnostic criteria assessor.
> ► Unblinding of the diagnostic criteria assessor.
> ► Quality of life questionnaires (4–17 years)—CDLQI and CHU-9D.
> ► Contact details (optional consent).
>
> **Medical record**
> ► Participant's diagnosis (reference standard).
> ► Age at diagnosis.
> ► Age at onset of symptoms.
> ► Skin biopsy result.
> ► Disease severity.
> ► Presence of psoriatic arthritis.
> ► Current skin treatments—topical, systemic, phototherapy.
> ► Clinical photographs (optional consent).
>
> CDLQI, Children's Dermatology Life Quality Index; CHU-9D, Child Health Utility 9D; DIPSOC, Developing DIagnostic criteria for PSOriasis in Children.

aged 4–17 years) and a diagnostic criteria assessment by a study investigator who is blinded to the participant's diagnosis (blinded to the reference standard). The two quality of life questionnaires are the Children's Dermatology Life Quality Index and the Child Health Utility 9D.

Each participant is offered a certificate, sticker and voucher to say thank you for taking part. Following the research visit, information is extracted from the medical record by an investigator who did not perform the assessment (blinded to the index test). Data to be extracted include the reference standard (diagnosis of skin disease), duration of disease, disease severity and current treatments. A summary of the data variables collected in the DIPSOC study is presented in box 2.

## Data management
Data are collected at the time of assessment and from the medical record. A number of steps have been taken to help ensure high-quality data collection. All DIPSOC study investigators undergo standardised training and receive a study manual to use as a practical guide when conducting the study. All DIPSOC diagnostic criteria assessors are trained using a PowerPoint presentation by EB-T (a clinical dermatologist with an interest in paediatric psoriasis) either face to face or by teleconference. Diagnostic criteria assessors come from both a dermatology and non-dermatology background. Understanding of the training material is checked using a short assessment based on clinical photographs. All assessors are required to achieve a minimum of 90% in the assessment prior to starting the study. The diagnostic criteria training manual is provided as a reference aid for investigators to use during their assessment.

The case report form includes guidance notes and was piloted to check for accuracy of completion. Quality of life is measured using validated measurement instruments. A data management process has been designed to minimise errors. All data monitoring is taking place centrally and data queries are checked with individual recruiting sites. Data checks are also built into the database design and all data for the primary objective will be entered twice.

## Consistency checks
To assess the interobserver variability in the diagnostic criteria assessment, the assessment will be conducted consecutively by two independent assessors in the first forty participants where two assessors are available.

To assess the variability in the reference standard for psoriasis, when optional consent is given, anonymised clinical photographs of cases taken as part of routine clinical care will be sent as anonymised case studies to the twelve consultant dermatologist principal investigators. The consultant dermatologists will be asked to score whether they agree or disagree with the diagnosis of psoriasis.

## Sample size and data analysis
The sample size is based on the primary objective. First, based on a 95% CI that the positive likelihood ratio (LR) is greater than 2 assuming a ratio of 1:1 cases to controls and an estimated sensitivity of 0.8 and specificity of 0.7 the sample size required is 74 cases and 74 controls.[27]

Second, transparent reporting of multivariable prediction models for individual prognosis or diagnosis (TRIPOD) has stated that there are no clear methods for calculating an adequate sample size. The guidance supports the current rule of thumb for sample size calculations of 10 events per variable.[28] As there are 16 diagnostic features in the consensus agreed diagnostic criteria, a sample size of 160 cases and 160 controls has been calculated.

Participant characteristics will be analysed using descriptive statistics. The diagnostic accuracy of the consensus agreed criteria will be calculated using sensitivity and specificity; 95% CIs will be presented. Multivariate logistic regression analysis will be used to develop the best predictive criteria using the DIPSOC data. The diagnostic features will be entered into the backward regression model. All minor criteria will be entered into the regression model and LR will be presented. Variation of diagnostic accuracy in different clinical contexts will be explored in stratified analysis for the following variables; age at the time of assessment, sex, assessor type and consultation type (new or follow-up).

The results of the multivariate analysis will be plotted on a receiver operator characteristic (ROC) curve and a coefficient threshold determined. The minimum threshold for the best predictive criteria has been set at 0.8 sensitivity and 0.8 specificity after consultation with the expert advisory group (detailed in the acknowledgements). The best predictive diagnostic criteria will be applied to the

study data and sensitivity and specificity calculated. The performance of the consensus agreed criteria and the best predictive model criteria will be compared using area under the ROC curve.

Interobserver variability and variability in the reference standard will be calculated using the kappa statistic. Further details on the analysis will be made public in the statistical analysis plan which will be shared on the DIPSOC website www.nottingham.ac.uk/go/dipsoc before the end of recruitment.

### Minimising bias

We have minimised selection bias by asking sites to approach all eligible cases and consecutive controls. All those approached but not recruited will be included in a screening log to demonstrate a non-selective approach. By minimising exclusion criteria, we aimed to design an inclusive study to support generalisation of the results.

The diagnostic criteria assessment will be undertaken by an investigator who is unaware of (blinded to) the dermatologist's diagnosis of the participant. Investigators are trained to focus on the presence or absence of each clinical feature. The study will test a prespecified scoring algorithm proposed through the eDelphi consensus study and a prespecified diagnostic threshold decided with the expert advisory group.

Bias in the reference standard will be minimised by ensuring all participants have a confirmed diagnosis by a dermatologist. As this is a case–control study, the reference standard will predate the index test. Variability in the reference standard will be examined using clinical photographs.

We have designed the study to include same day recruitment directly from clinic, therefore, the time between the reference standard and index test for most participants will be short. All participants receive the same reference standard (a dermatologist's diagnosis), therefore, complete verification will be achieved. All participants will be included within the analysis and a complete data set sensitivity analysis is planned. Data are extracted from the medical record by an investigator who did not undertake the diagnostic criteria assessment (blinded to the index test).

### Patient and public involvement

The patient and public involvement aim in this study was to inform our understanding of the importance of diagnosis to patients, to make sure the study design was patient centred, to ensure the participant facing documents were what patients wanted and to inform dissemination. A patient advisor (CH) has been involved from the beginning of the project and is a study coauthor. PAPAA, a UK patient association, is a supporting organisation. We have also met the Young Person's Advisory Group (YPAG) for Research Nottingham and patients in paediatric dermatology clinics.

The research questions were developed and informed by the patient association prioritisation work, discussion

---

**Box 3  Suggestions from patient and public involvement work that have informed the DIPSOC study**

**Study design.**
- ► Include on the day recruitment and the option to attend for a separate research visit.
- ► Invite participants by letter in advance of their clinic appointment.

**Participant information sheets**
- ► Change the format to a leaflet or booklet.
- ► Colourful boxes around the text and different colours for different sections.
- ► Emphasise confidentiality and the assessment will take place in a private space.
- ► Include photographs of the research team.
- ► Do not include photographs of psoriasis.
- ► Provide electronic versions of the information sheets on a website.

Create a distinctive logo for the study
Provide a colouring-in sheet.
Give a certificate and sticker at the end of the research visit.

---

with our patient advisor and the YPAG. We discussed what diagnosis means to young people and the importance of being able to give a name to a disease. Important suggestions from these groups that have informed the study are presented in box 3. The above groups will guide dissemination to patient communities.

### Substudy

The objective of the substudy is to assess the performance of the best predictive diagnostic criteria to identify psoriasis in children/young people currently diagnosed with indeterminate disease. Children and young people with possible or indeterminate psoriasis will be recruited alongside the main study to the nested substudy. The eligibility criteria and research visit are otherwise identical to the main study. No control participant is required. Children/young people in the substudy will, if consent is provided, be sent a questionnaire 2 years after the last participant is recruited. The questionnaire will ask about their skin disease and whether the diagnosis has changed. The substudy data will be used to calculate the sensitivity and specificity of the best predictive criteria in identifying children and young people previously diagnosed with indeterminate psoriasis who go on to be diagnosed with psoriasis.

### ETHICS AND DISSEMINATION
### Ethics

The four principles of biomedical ethics were considered in the study design and documentation. The purpose, aims and details of taking part in the study are explained in the participant information sheets. It is explained that taking part is voluntary and not taking part will have no effect on the patient's medical care. Informed consent is necessary before any part of the study is completed. It is also explained that taking part

in the study will have no direct medical benefit for the patient, but may help the diagnosis of other children or young people in the future. The study is non-interventional and non-therapeutic. All study investigators are required to be Good Clinical Practice trained.

## Dissemination

Dissemination will be guided by stakeholders; patients, children and young people, dermatologists, primary care and paediatric rheumatologists. The aim is to publish the study results in a high-quality peer-reviewed journal and present the findings at international academic meetings. The results will also be shared through social media and the supporting patient association (PAPAA).

## DISCUSSION

DIPSOC is a development study and the first in a series of studies needed to develop, test and validate the diagnostic accuracy of criteria for psoriasis in children/young people. The nested substudy will be important to investigate whether the criteria can help identify children with psoriasis at an indeterminate stage, before their skin disease may have fully evolved.

The development and introduction of diagnostic criteria for psoriasis in children/young people has the potential to improve the early and accurate recognition of psoriasis and juvenile psoriatic arthritis, prompt referral for specialist assessment and monitoring, standardise clinical research to enable meta-analysis of data and support case finding in new epidemiological studies. The utility of diagnostic criteria will, therefore, be in primary and secondary care as well as clinical research.

## Limitations

DIPSOC has been designed to ensure a high-quality diagnostic study, but there are some important limitations. A case–control study design is likely to overestimate the diagnostic accuracy of the criteria. DIPSOC is a development study, and therefore, this study design is appropriate and feasible for this early stage of testing. In the future, further diagnostic cohort studies are needed to test, potentially improve, and validate the resulting criteria in the setting and population they are intended to be used. Another limitation of the study design is spectrum bias. Participants recruited from paediatric dermatology clinics are likely to have more severe and persistent disease (ie, a different clinical presentation) compared with children/young people in the community who are managed by general practitioners. This spectrum bias may lead to an overestimation of the diagnostic accuracy because participants may have more obvious disease. DIPSOC recruits both new and follow-up (incident and prevalent) patients. This will include participants currently on treatment who's skin rashes may have changed since starting treatment. However, paediatric dermatology clinics are a feasible setting to recruit the required sample size and

obtain a reference standard to ensure complete verification. DIPSOC does not include external validation of the best predictive criteria and this will need to be undertaken in separate studies once the diagnostic criteria have been developed.

## Study progress

Twelve UK centres are open for recruitment. These centres are Nottingham, Barts London, Middlesbrough, Cambridge, Sheffield, Coventry, Glasgow, Dorchester, Oxford, St George's London, Plymouth and Cardiff. The first participant was recruited in October 2017 and the study is due to finish recruiting on August 2019. We are currently in the data collection phase.

**Acknowledgements** We would like to thank the following experts for their advice and guidance on the study design and analysis plan. Hywel C. Williams, Centre of Evidence Based Dermatology, University of Nottingham, Nottingham, UK. Luigi Naldi, Centro Studi GISED, Bergamo, Italy. Miriam Santer, Primary Care and Population Sciences, University of Southampton, Southampton, UK. Nick Francis, Division of Population Medicine, Cardiff University, Cardiff, UK. Phillip Helliwell, Leeds Institute of Rheumatic and Musculoskeletal Medicine, University of Leeds, Leeds, UK. Matthew Grainge, Division of Epidemiology and Public Health, School of Medicine, University of Nottingham, Nottingham, UK. Sinead Langan, London School of Hygiene and Tropical Medicine, London, UK. Test Evaluation Research Group at the University of Birmingham led by Jon Deeks. We would like to thank the following for their advice and guidance on developing the diagnostic criteria assessment training manual and case report form. Jonathan Batchelor, Centre of Evidence Based Dermatology, University of Nottingham, Nottingham, UK. Karen Harman, Centre of Evidence Based Dermatology, University of Nottingham, Nottingham, UK. Rosalind Simpson, Centre of Evidence Based Dermatology, University of Nottingham, Nottingham, UK. Joanne Chalmers, Centre of Evidence Based Dermatology, University of Nottingham, Nottingham, UK. Joanne Llewellyn, Centre of Evidence Based Dermatology, University of Nottingham, Nottingham, UK. Faye Shelton and Emma Smith, who are no longer affiliated with the Centre of Evidence Based Dermatology. We would like to thank the Psoriasis and Psoriatic Arthritis Alliance for their support of the research study.

**Contributors** EB-T is an National Institute for Health Research (NIHR) Doctoral Research Fellow and study coordinator for the DIPSOC study. RM is Consultant Adult and Paediatric Dermatologist and is the study's medical expert. SG is an Assistant Professor and medical statistician. TN is a Consultant Dermatologist and Professor of Dermato-epidemiology, providing medical and methodological expertise. CH is a patient advisor. KST is a Professor of Applied Dermatology and Chief Investigator for the study. EB-T led on writing the protocol. RM, SG, TN, CH and KST commented and approved the protocol. SG and TN provided expertise on the statistical analysis plan. All authors read and approved this manuscript.

**Funding** Esther Burden-Teh is funded through an NIHR Doctoral Research Fellowship (DRF-2016-09-083).

**Disclaimer** The views expressed are those of the author(s) and not necessarily those of the NHS, the NIHR or the Department of Health and Social Care. The funder reviewed the study design and will approve the manuscript prior to publication.

**Competing interests** None declared.

**Patient consent for publication** Not required.

**Ethics approval** Health Regulatory Authority (HRA) and National Health Service Research Ethics Committee (NHS REC) approvals were granted in February 2017 (REC Ref: 17/EM/0035). The study follows the Declaration of Helsinki.

**Provenance and peer review** Not commissioned; externally peer reviewed.

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
