## [Reviewer comments · BMJ Open]

ARTICLE DETAILS

TITLE (PROVISIONAL)	Protocol for a case-control diagnostic accuracy study to develop diagnostic criteria for psoriasis in children (DIPSOC study): a multicentre study recruiting in UK paediatric dermatology clinics.
AUTHORS	Burden-Teh, Esther; Murphy, Ruth; Gran, Sonia; Nijsten, Tamar; Hughes, Carolyn; Thomas, Kim

VERSION 1 - REVIEW

REVIEWER	Álvaro Machado Department of Dermatology, Centro Hospitalar do Porto, Porto, Portugal
REVIEW RETURNED	28-Mar-2019

GENERAL COMMENTS	The authors presented an interesting protocol which could have impact in clinical practice. In fact, psoriasis in children often is a diagnostic challenge and this could represent an useful tool to all dermatologists.
--

REVIEWER	Kristian Reich Center for Translational Research in Inflammatory Skin Diseases, Institute for Health Care Research in Dermatology and Nursing, University Medical Center Hamburg-Eppendorf, Hamburg, Germany
REVIEW RETURNED	31-Mar-2019

GENERAL COMMENTS	Straight forward study design and important subject. There are a few spelling mistakes. Question: How would PRP type IV be distinguished? Is there biopsy proof in unclear cases? Can the results be used for AI learning of RW cases at a later stage?
---

REVIEWER	Marieke Seyger Marieke M.B. Seyger, MD, PhD Associate Professor of Dermatology Radboudumc Principal Clinician marieke.seyger@radboudumc.nl T +31 (0)24 361 37 24
-----------------	---

	Radboud university medical center Department of Dermatology, Center of Excellence for Pediatric Psoriasis and Psoriasis P.O. Box 9101, 6500 HB Nijmegen (370), The Netherlands René Descartesdreef 1 (route 370) www.radboudumc.nl
REVIEW RETURNED	19-May-2019

GENERAL COMMENTS	Indeed, the diagnosis of psoriasis in children and young people may be missed by non-dermatologists, and especially for general practitioners (GPs) the differentiation between atopic dermatitis (AD) and psoriasis can be a challenge. The primary objective is to test the diagnostic accuracy of the consensus agreed diagnostic criteria and to develop the best predictive diagnostic criteria for pediatric psoriasis. Controls in this study have a confirmed diagnosis of a scaly inflammatory rash by a dermatologist. The decision as to whether a participant's skin disease meets the eligibility criteria is made by the patients dermatologist. What are the exact eligibility criteria? It is not stated that the rash in the control group should be erythematous (ichthyosis is mentioned as an example, which is often not erythematous), although 2 of 3 major diagnostic criteria for psoriasis are the presence of erythematous plaques. Shouldn't that (erythematous) be one of the eligibility criteria for inclusion as control? By allowing inclusion of many (not pre-specified) diagnoses as controls in this case control study it is even more likely to overestimate the diagnostic accuracy of the criteria. Wouldn't it be better to pre specify a list of erythematous scaly diagnoses to be included as controls? Because the distinction between AD and psoriasis in childhood is most challenging, isn't it even better to include a majority of patients with AD as controls? What is the educational background of the diagnostic criteria assessors? Could you please specify? If these assessors are for example trainees in dermatology the accuracy is likely to be higher than eg a rheumatology nurse. Selection of assessors could bias the results. How will the authors correct for the influence of the powerpoint training on the accuracy of the diagnostic criteria? Obviously, training of clinical appearance of pediatric psoriasis and creating awareness of this diagnosis improves diagnostic accuracy. How will the authors differentiate between the influence of the training itself on accuracy vs the accuracy of the criteria? Included cases have a confirmed diagnosis of plaque psoriasis. The authors state that guttate psoriasis can be included as a subtype or presentation of plaque psoriasis. This puzzles me, as literature defines guttate psoriasis as a separate entity (eg Cochrane Database Syst Rev. 2019 Apr 8;4). Box 1 describes the diagnostic criteria agreed to be important for plaque psoriasis. Therefore it seems better to exclude children with guttate psoriasis.
---

REVIEWER	G.E. van der Kraaij Amsterdam UMC, The Netherlands
REVIEW RETURNED	24-May-2019

GENERAL COMMENTS	It is an important project and well thought out protocol.
---

	No comments to be made. I hope for a prosperous data collection and look forward to the results!
--	--

VERSION 1 – AUTHOR RESPONSE

Reviewer(s) Reports:

Reviewer: 1

Reviewer Name: Álvaro Machado

Institution and Country: Department of Dermatology, Centro Hospitalar do Porto, Porto, Portugal
Please state any competing interests or state 'None declared': None declared.

Please leave your comments for the authors below The authors presented an interesting protocol which could have impact in clinical practice.

In fact, psoriasis in children often is a diagnostic challenge and this could represent an useful tool to all dermatologists.

Thank you very much for appreciating the value of this study.

Reviewer: 2

Reviewer Name: Kristian Reich

Institution and Country: Center for Translational Research in Inflammatory Skin Diseases, Institute for Health Care Research in Dermatology and Nursing, University Medical Center Hamburg-Eppendorf, Hamburg, Germany Please state any competing interests or state 'None declared': None

Please leave your comments for the authors below Straight forward study design and important subject.

There are a few spelling mistakes.

Thank you for letting us know that there are some spelling mistakes, we will rectify these.

Question: How would PRP type IV be distinguished? Is there biopsy proof in unclear cases? Can the results be used for AI learning of RW cases at a later stage?

Thank you. The eligibility criteria for both cases and controls require patients to have a clinical diagnosis, which may be supported by a biopsy. Therefore, if required as part of the patient's diagnostic work up, a skin biopsy may be performed prior to the patient being recruited to the study. PRP type IV could be distinguished in this way. It is an interesting suggestion about utilising the data for AI learning. As part of the consent process, participants/parents were consented for the anonymised data to be used by other researchers. Therefore, data from DIPSOC could be used to inform AI learning in the future.

Reviewer: 3

Reviewer Name: Marieke Seyger

Institution and Country: Marieke M.B. Seyger, MD, PhD, Associate Professor of Dermatology, Radboud university medical center, Nijmegen, The Netherlands Please state any competing interests or state 'None declared': none declared

Please leave your comments for the authors below Indeed, the diagnosis of psoriasis in children and young people may be missed by non-dermatologists, and especially for general practitioners (GPs) the differentiation between atopic dermatitis (AD) and psoriasis can be a challenge.

The primary objective is to test the diagnostic accuracy of the consensus agreed diagnostic criteria and to develop the best predictive diagnostic criteria for pediatric psoriasis.

Controls in this study have a confirmed diagnosis of a scaly inflammatory rash by a dermatologist. The decision as to whether a participant's skin disease meets the eligibility criteria is made by the patients dermatologist. What are the exact eligibility criteria? It is not stated that the rash in the control group should be erythematous (ichthyosis is mentioned as an example, which is often not erythematous), although 2 of 3 major diagnostic criteria for psoriasis are the presence of erythematous plaques. Shouldn't that (erythematous) be one of the eligibility criteria for inclusion as control? By allowing inclusion of many (not pre-specified) diagnoses as controls in this case control study it is even more likely to overestimate the diagnostic accuracy of the criteria. Wouldn't it be better to pre specify a list of erythematous scaly diagnoses to be included as controls? Because the distinction between AD and psoriasis in childhood is most challenging, isn't it even better to include a majority of patients with AD as controls?

Thank you for your helpful comments. As this is a protocol for an ongoing study, it is not possible to change the inclusion criteria at this stage but we will bear in mind your comments when interpreting the results.

The study's eligibility criteria are included in manuscript subsection 'Inclusion criteria'. As per the QUADAS-2 critical appraisal tool, exclusions were kept to a minimum. The criteria for a control requires the dermatologist's diagnosis of a scaly inflammatory rash, but erythema wasn't specified. The aim for defining the control population was to identify the population from which cases would be identified, and a scaly inflammatory rash was decided to best describe this. A pre-specified list would have introduced exclusions, and at the time of recruitment the skin changes may not have matched the description of a scaly inflammatory rash (for example, the scaly component may have resolved). It

is likely, due to the prevalence of eczema in UK paediatric dermatology clinics, that the majority (>80%) of control participants are likely to have atopic dermatitis.

The following is included in the manuscript under the section minimising bias: By minimising exclusion criteria we aimed to design an inclusive study to support generalisation of the results.

What is the educational background of the diagnostic criteria assessors? Could you please specify? If these assessors are for example trainees in dermatology the accuracy is likely to be higher than eg a rheumatology nurse. Selection of assessors could bias the results.

There are two types of assessors: (i) dermatology trained (derm consultant, paed consultant, derm register and derm nurse), and (ii) dermatology untrained (other doctor, non-derm nurse, other investigator). Included in the statistical analysis plan, is a planned stratified analysis to determine the diagnostic accuracy for these two groups separately and compare results.

The following has been added to the manuscript under the section titled statistic analysis plan: Variation of diagnostic accuracy in different clinical contexts will be explored in stratified analysis for the following variables; age at the time of assessment, sex, assessor type and consultation type (new or follow-up).

How will the authors correct for the influence of the powerpoint training on the accuracy of the diagnostic criteria? Obviously, training of clinical appearance of pediatric psoriasis and creating awareness of this diagnosis improves diagnostic accuracy. How will the authors differentiate between the influence of the training itself on accuracy vs the accuracy of the criteria?

Thank you for your helpful comment, we will mention this when interpreting the results in the Discussion section of our paper. Standardised training was necessary to ensure assessors approached assessment of skin changes for the presence or absence of diagnostic criteria in a standardised way. Training is therefore part of the accurate application of the diagnostic criteria.

Included cases have a confirmed diagnosis of plaque psoriasis. The authors state that guttate psoriasis can be included as a subtype or presentation of plaque psoriasis. This puzzles me, as literature defines guttate psoriasis as a separate entity (eg Cochrane Database Syst Rev. 2019 Apr 8;4). Box 1 describes the diagnostic criteria agreed to be important for plaque psoriasis. Therefore it seems better to exclude children with guttate psoriasis.

Thank you for your comment. Plaque psoriasis was defined as a diagnosis of psoriasis where plaques are the dominant feature. Guttate psoriasis fulfils this definition. At the time of the eDelphi, the eDelphi International Psoriasis Council participants agreed that guttate psoriasis could be included within this broad definition. The rationale for this was that it can be difficult to decide where guttate ends and

chronic plaque psoriasis begins in a patient that is transitioning from one to the other. As this is an ongoing study it is not possible to change the eligibility criteria, but we will mention this in the Discussion section of our results paper.

The following has been added to the section participant selection: The decision to include guttate psoriasis under the broad description of plaque psoriasis was agreed with the International Psoriasis Council.

Reviewer: 4

Reviewer Name: G.E. van der Kraaij

Institution and Country: Amsterdam UMC, The Netherlands

Please state any competing interests or state 'None declared': none declared

Please leave your comments for the authors below It is an important project and well thought out protocol.

No comments to be made. I hope for a prosperous data collection and look forward to the results!

Thank you very much for appreciating the value of our work.

VERSION 2 – REVIEW

REVIEWER	Marieke Seyger Radboud university medical center, Nijmegen, The Netherlands
REVIEW RETURNED	06-Jul-2019

GENERAL COMMENTS	The authors have addressed all questions well. Thank you for the opportunity to review. I am looking forward to the results!
--